

# Wave–Current Interactions in a Wind-jet Region

Laura Ràfols[a,b], Manel Grifoll[a], and Manuel Espino[a]

[a]Maritime Engineering Laboratory (LIM-UPC), Polytechnic University of Catalonia (BarcelonaTech), C/ Jordi Girona 1-3 Edif. D1, 08034 Barcelona, Spain
[b]Meteorological Service of Catalonia (SMC), C/ Berlin 38-48 4a, 08029 Barcelona, Spain

**Correspondence:** Laura Ràfols (laura.rafols@upc.edu)

**Abstract.** Wave–Current Interactions (WCIs) are investigated. The study area is located at the northern margin of the Ebro Shelf (northwestern Mediterranean Sea), where episodes of strong cross-shelf wind (wind jets) occur. The aim of this study is to validate the implemented coupled system and investigate the impact of WCIs on the hydrodynamics of a wind-jet region. The Coupled Ocean–Atmosphere–Wave–Sediment Transport (COAWST) modeling system, which use Regional Ocean Model

System (ROMS) and Simulating WAves Nearshore (SWAN) models, is used in a high-resolution domain (350 m). Results from uncoupled numerical models are compared with a two-way coupling simulation. The results do not show substantial differences in the water current field between the coupled and the uncoupled runs. The main effect observed when the waves are considered is in the water column stratification, due to the turbulent kinetic energy injection and the enhanced surface stress, leading a larger mixed-layer depth. Additionally, when the water currents are considered, the agreement of the modeled wave

period significantly improves and the wave energy (and thus the significant wave height) decreases when the current flows in the same direction as the waves propagate.

# 1   Introduction

During the last decade, several water circulation models have been developed including the wind-waves induced currents. There are two different formulations to include the so-called wave effects on currents (WEC) in the three-dimensional primitive equations: by means of the radiation stress gradient (Mellor, 2011) and with the vortex force (VF) formalism (Uchiyama et al., 2010; Kumar et al., 2012). The VF formalism separates the conservative and non-conservative contributions in the momentum

balance equations, which allows one to evaluate flow fields within both inner shelf and surf zone environments (Kumar et al., 2012).

From a modeling perspective, several circulation and wave models have been coupled in order to consider the wave–current interactions (WCIs). For instance, Xie et al. (2001) coupled the 3D ocean model POM with the WAM wave model and found



that wind waves can significantly affect coastal ocean currents both at the surface and near the seabed. Osuna and Wolf (2005) implemented the coupling between the circulation Proudman Oceanographic Laboratory Coastal-Ocean Modeling System (POLCOMS) and the WAM model in the Irish Sea. This system was then modified by Bolaños et al. (2011), who included three-dimensional interactions following Mellor (2003, 2005) and applied the coupled model system to the Mediterranean Sea.

Tang et al. (2007) implemented the WCI in a 3D ocean model (Princeton Ocean Model, POM) and a spectral wave model (WAVEWATCH III), based on Jenkins (1987) formulation, and evaluated the model by comparison with surface velocity data derived from surface drifters. McWilliams et al. (2004) developed a multi-scale asymptotic theory for the evolution and interaction of currents and surface gravity waves of finite depth, which was then implemented and extended for applications within the surf zone in the UCLA ROMS model by Uchiyama et al. (2010). Warner et al. (2008b) used the Model Coupling

Toolkit (MCT) to couple the ocean circulation model Regional Ocean Model System (ROMS) and the surface wave model Simulating WAves Nearshore (SWAN) and included nearshore processes, such as radiation-stress terms based on Mellor (2003, 2005) and a surface roller model (Svendsen, 1984; Svendsen et al., 2002). This system was then further developed by Warner et al. (2010) to include one-way grid refinement in the oceanic and wave models, coupling to an atmospheric model in order to include effects of sea surface temperature and waves, and to provide interpolation mechanisms to allow the models to compute

on different grids. The resulting system is known as the Coupled Ocean–Atmosphere–Wave–Sediment Transport (COAWST) modeling system. Then, Kumar et al. (2012) implemented the VF formalism into the COAWST modeling system, with some modifications to the method of Uchiyama et al. (2010).

The north Ebro Shelf (NW Mediterranean Sea) is a region characterized by northwestern (NW) winds that are channeled through the Ebro Valley and which result in cross-shelf wind jets when they reach the sea. This region is very interesting from a

meteo-oceanographic point of view because multiple processes take place, such as bimodal wave spectra and the development of a two-layer cross-shelf flow. Some authors have investigated the circulation patterns (Grifoll et al., 2015; Ràfols et al., 2017a) and the wave field (Bolaños-Sanchez et al., 2007; Grifoll et al., 2016; Ràfols et al., 2017b) during these NW wind-jet events but less efforts have been made at investigating the WCI in the region. Due to the limited observational data, in order to study the wind-jet induced dynamics of the region, the use of numerical models is required. However, at the same time, this makes

the investigation rather challenging and forces a rather qualitative analysis based on the modeled physical processes reliability. The purpose of this study is to validate the implemented coupled system and investigate the wave effects on the circulation and the current effects on the wave field at the continental shelf during a wind-jet event. With this aim, results from uncoupled models are compared with the outputs from a two-way coupled numerical model. The selected study period is from March 15 2014 to May 15 2014 because it contains four wind-jet episodes. Additionally, this period has been previously used to validate

numerical models in the study region and the wind-wave characterization (Ràfols et al., 2017b), and water shelf circulation (Ràfols et al., 2017a) was investigated by combining numerical efforts and in situ observations.

This work is organized as follows. In Section 2 the study area and the methods used in this work are presented. The results are shown in Section 3, discerning between the effects of waves on currents and the effects of currents on waves. A discussion of the results can be found in Section 4, and the main conclusions of the work are highlighted in Section 5.



## 2    Study area and methodology

### 2.1    Study area

The north Ebro Shelf is located at the southern part of the Catalan coast, at 40.4°–41.1° N and 0.4°–1.3° E (see Fig. 1). The shelf of this region is characterized by the transition from a narrow shelf (∼10km) at its northern margin to a broader shelf (∼60 km) towards the south.

    The most characteristic wind of the region is the northwesterly wind (mistral), which is channeled through the Ebro Valley resulting in a cross-shelf wind jet when it reaches the sea. This wind jet is related to the presence of a high-pressure area over the Iberian Peninsula and a low-pressure area over the Mediterranean Sea. Thus, it is more common during autumn and winter (Grifoll et al., 2015), when large atmospheric pressure gradients occur. The predominant regional current is the quasi-permanent slope current known as the Northern Current, which is an entity flowing along the continental slope (Millot, 1999) that can be modified by mesoscale events such as current meandering or eddies (Font et al., 1995). The water circulation in the inner and mid-shelf presents strong temporal and spatial variability due to the strong gradients in the bathymetry and wind field associated with wind-jet episodes (Grifoll et al., 2015; Ràfols et al., 2017a). The wave climate at the Ebro Delta is characterized by the predominance of NW winds (which coincides with the predominance of NW winds), although there are also significant storms from the east and south. These storms tend to develop a bimodal directional spectrum due to the coexistence of wind waves and swell waves (Sánchez-Arcilla et al., 2008; Ràfols et al., 2017b). Local wind waves (sea system) show a broadband spectrum with a high variety of frequencies associated with irregular sea states. In contrast, waves generated far away (swell system) present a narrowband spectrum with a frequency range with less associated energy. Then, when the sea and swell systems exist at the same time, bimodal spectra occur (Ràfols et al., 2017b).

### 2.2    Data

For validation purposes, oceanographic and coastal meteorological measurements from Puertos del Estado (http://www.puertos.es) are used. Specifically, data obtained from a coastal wave buoy, a deep-water buoy and an high-frequency (HF) radar. The locations are shown in Fig. 1, jointly with the bathymetry and the numerical domains.

    The coastal wave buoy (CB) is a Triaxys buoy located at 41.07° N, 1.19° E at 15 m depth, deployed in November 1992. It provides significant wave height, peak period, nautical direction and directional wave spectra, among other data. The deep-water buoy (DB), an ocean Seawatch buoy located at 40.68° N, 1.47° E at 688 m depth was deployed in August 2004. This buoy measures water velocity and water temperature at the sub-surface (nominal depth of 3 m), wind vectors at 3 m above the sea surface, significant wave height, peak period, nautical direction and directional wave spectra, among other parameters. In order to be able to compare the measured wind data at 3 m height with the modeled data at 10 m height, the modeled data have been extrapolated from 10 m to 3 m using a logarithmic profile (see Appendix A).

    The HF radar system used in this work is a CODAR SeaSonde standard-range system composed of three remote shelf-based sites that became operational in December 2013. Each site comprises a transmitter–receiver antenna that operates at a nominal frequency of 13.5 MHz with a 90 KHz bandwidth. The system provides hourly measurements of the current velocities in the





**(a)**
**(b)**

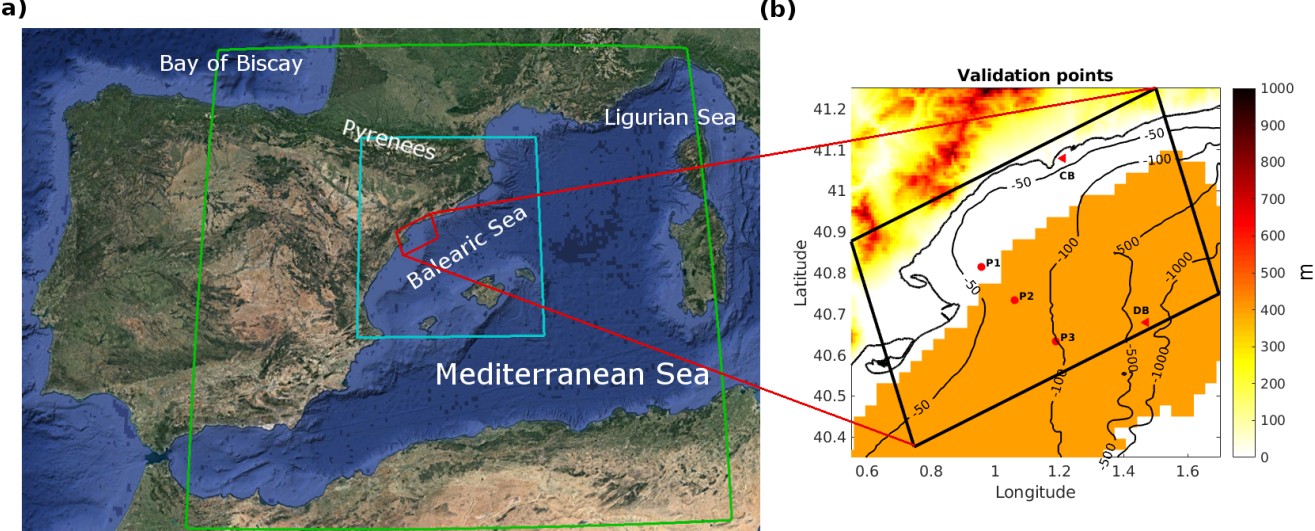

**Figure 1.** Study area. (a) NW Mediterranean Sea and numerical domains: 15 km resolution domain for the SWAN model (green), 3 km resolution domain for the SWAN model (blue) and 350 m resolution coastal domain for the ROMS and SWAN models (red). (b) Orography (in m), coastal domain, buoy locations (red triangles; CB and DB), points where the numerical results are examined in detail (red dots: P1, P2 and P3) and HF radar coverage area (in orange).

top meter of the water column with a horizontal resolution of 3 km and a cut-off filter of 100 cm/s. More information about the system is available in Lorente et al. (2015).

## 2.3 Numerical models

### 2.3.1 COAWST modeling system description

5 The COAWST Modeling System (Warner et al., 2010) has been widely used by many authors to investigate the WCI (Olabarrieta et al., 2011; Renault et al., 2012; Benetazzo et al., 2013; Rong et al., 2014; Grifoll et al., 2014; Bruneau and Toumi, 2016, among others). In this study, the COAWST modeling system is used to perform the uncoupled ROMS and SWAN model simulations and the two-ways coupling run.

The SWAN model is a third-generation numerical wave model that computes random, short-crested waves in coastal regions 10 with shallow water and ambient currents (Booij et al., 1999). It is based on the action balance equation in terms of action density N (Bretherton and Garrett, 1968) with sources and sinks and incorporates state-of-the-art formulations of wave–wave interactions and the processes of wave generation and dissipation.

The ROMS model is a split-explicit, free-surface, terrain-following, primitive equations oceanic model that solves the 3D Reynolds-Averaged Navier–Stokes equations using the hydrostatic and Boussinesq assumptions (Shchepetkin and McWilliams,



2005; Haidvogel et al., 2008). The model uses finite-difference approximations on a terrain-following vertical coordinate (sigma coordinate) and on a horizontal curvilinear Arakawa C grid.

The Model Coupling Toolkit (MCT; Larson et al., 2004; Jacob et al., 2004) is a Fortran90 program that works with the MPI protocol. It allows the transmission and transformation of various distributed data between component models using a parallel

coupled approach. When the models are initialized, each model decomposes its own domain into different sections, which are distributed to processors. On each processor, each grid section initializes into MCT and a global map of the distribution of the segments is computed. Each segment also initializes an attribute vector that contains the fields to be exchanged and establishes a router to provide an exchange pathway between model components. While the simulation is run, the models reach a synchronization point, fill the attribute vectors with data and exchange fields. Further details are described in Warner et al.

(2008a).

### 2.3.2 System set-up

Three different runs have been performed in this work (see Fig. 2): one with the ROMS model uncoupled, one with the SWAN model uncoupled and, finally, one with the ROMS and SWAN models two-way coupled.

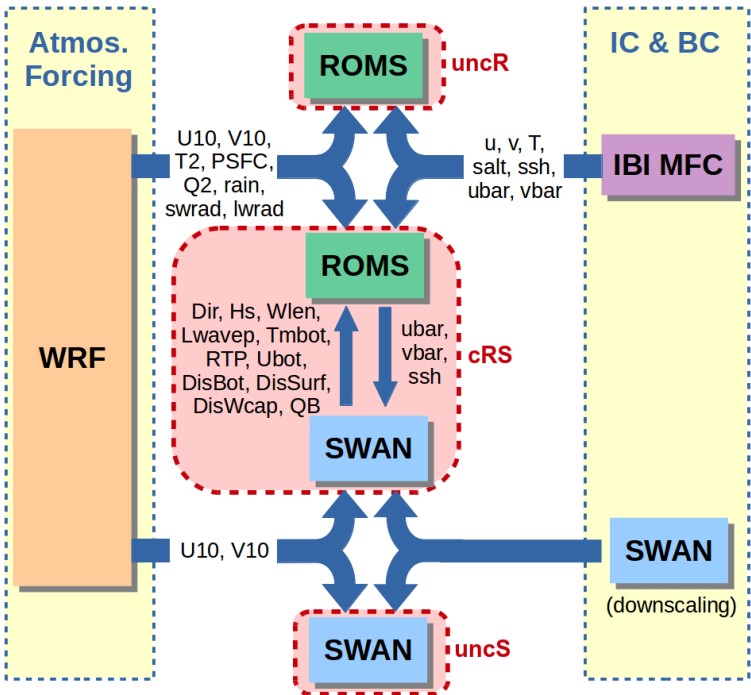

**Figure 2.** Configuration of setup run. In red, the name given to each configuration.

The numerical domain has a horizontal resolution of 350 m and, in the ROMS case, a vertical resolution of 20 sigma levels.

The bathymetry introduced in the models has a grid resolution of 0.0083° and was obtained from General Bathymetric Chart





of the Oceans (GEBCO; www.gebco.net). Both SWAN and ROMS models are forced with hourly atmospheric data from a previous WRF (Weather Research and Forecasting) model run provided by the SMC (Servei Meteorològic de Catalunya) that has a spatial resolution of 3 km.

In order to generate the boundary conditions for the SWAN model, a downscaling technique has been used. The entire system consists of three nested domains (see Fig. 1a). The largest one covers the western Mediterranean Sea with a spatial resolution of 15 km and provides boundary conditions to a second-level domain. The latter covers the Balearic Sea with a spatial resolution of 3 km and provides boundary conditions to the smaller domain, which has a horizontal resolution of 350 m. This study is focused on this last domain. The WRF model provided by the SMC provides the 10-m surface winds ($U10$, $V10$) forcing and the initial conditions have been obtained running the model in stationary mode.

In the SWAN model, non-stationary conditions, spherical coordinates and nautical convention have been selected. The wave growth by wind is computed with a sum of a linear term and an exponential term. For the linear growth, the expression from Cavaleri and Malanotte-Rizzoli (1981) is used, and for the exponential growth, the expression and coefficients from Komen et al. (1984) are used. The nonlinear quadruplet wave interactions are integrated by a fully explicit computation of the nonlinear transfer with the Discrete Interaction Approximation (DIA; proposed by Hasselmann et al., 1985) per sweep (using default coefficients). For the whitecapping, the Komen et al. (1984) formulation is used with $C_{ds} = 2.36 \times 10^{-5}$ , $\delta = 1$ and $p = 4$. Finally, the JONSWAP (Hasselmann et al., 1973) bottom friction formulation is added with the default coefficients. The spectrum is discretized with a constant relative frequency resolution of $\Delta f = 1.1$ (logarithmic distribution) and a constant directional resolution of $\Delta\theta = 10°$. The discrete frequencies are defined between 0.01 Hz and 1 Hz. Above the high-frequency cutoff, a diagnostic tail $f^{-4}$ is added.

The initial and boundary conditions for the ROMS model are taken from the Iberian Biscay Irish – Monitoring and Forecasting Centre (IBI-MFC) product. This product (http://marine.copernicus.eu/) includes all main forcings (i.e. tidal forcing, high-frequency atmospheric forcing, fresh water river discharge, etc.) and is based on a (eddy-resolving) NEMO model application run at 1/36° horizontal resolution. The outputs provided by the IBI-MFC used in our numerical model are 3D daily means of temperature ($T$), salinity ($salt$) zonal velocity ($u$), meridional velocity ($v$) and 2D (surface) hourly means of sea surface height ($ssh$) and barotropic currents ($ubar$, $vbar$). The WRF model provided by the SMC provides the atmospheric forcing fields for the ROMS model, which include 10 m surface winds ($U10$, $V10$), atmospheric pressure ($PSFC$), relative humidity ($Q2$), atmospheric surface temperature ($T2$), precipitation ($rain$) and shortwave ($swrad$) and longwave ($lwrad$) net heat fluxes to the ocean model. The model uses these parameters in the COARE algorithm (Fairall et al., 1996) to compute ocean surface stresses and ocean surface net heat fluxes.

The ROMS model implementation includes a generic length-scale turbulent vertical mixing scheme with the $k-\epsilon$ parametrization, a logarithmic profile for the bottom boundary layer with a bottom roughness of 0.005 m and horizontal mixing terms in geopotential surfaces. The Ebro River discharge is characterized with data from the Automatic Hydrologic Information System of the Ebro River basin (owned by the Confederación Hidrográfica del Ebro, www.chebro.es). The data used to force the numerical model consist of daily measurements of river runoff and temperature.



In the two-way coupled run, the WEC are implemented using a coupling time step of 20 min. The wave model provides wave direction ($Dir$), significant wave height ($Hs$), wave length ($Wlen$), peak wave length ($Lwavep$), surface and bottom periods ($RTP$, $Tmbot$), bottom orbital velocity ($Ubot$), wave energy dissipation ($DisBot$, $DisSurf$, $DisWcap$) and percent wave breaking ($QB$) to the ocean model. These parameters are used by the ocean model in four different mechanisms:

– To compute enhanced bottom stresses due to the effect of turbulence in the wave boundary layer by means of the SSW (Sherwood / Signell / Warner) implementation of Madsen (1994) bottom boundary layer formulation.

   – To compute enhanced surface stresses (SStr) due to changes in the surface roughness $z_0$. In contrast to the COARE algorithm used in the uncoupled ROMS run, now the Taylor and Yelland (2001) sea surface roughness closure model, which is sea-state dependent, is used. Now the $z_0$ is derived from $\frac{z_0}{Hs} = 1200(Hs/Lp)^{4.5}$, where $Lp$ is the peak wave
length.

   – To inject turbulent kinetic energy (TKE) at the surface due to breaking waves. It is introduced as a surface flux of turbulence kinetic energy in the generic length scale method (Warner et al., 2005).

   – To include the wave forces using the VF formalism (Uchiyama et al., 2010; Kumar et al., 2012, ; see Section 2.4).

   The wave model receives currents ($u_s$, $v_s$) and sea surface height ($ssh$) from the ocean model. The surface currents ($u_s$,
$v_s$) were computed taking into account the vertical distribution of the current profile using the formulation presented by Kirby and Chen (1989), which integrates the near-surface velocity over a depth controlled by the wave number. The presence of an ambient current may change the amplitude (e.g. due to an energy transfer between waves and currents), the frequency (due to the Doppler shift) and the direction (due to current-induced refraction) of the waves. In this sense, the ocean currents modify the wind speed forcing with $S = f(U_{wind} - u_s; V_{wind} - v_s)$, the wave celerity using the modified group velocities $c_x = c_{gx} + u_s$,
$c_y = c_{gy} + v_s$ and the wave number (derived from the Doppler shift effect; see Holthuijsen (2008), Appendix D).

### 2.4 Momentum balance description

The cross-shelf momentum balance is used to analyze the wave effects on the circulation over the continental shelf. The simplified equations for the VF approach can be obtained after removing the curvilinear terms, body forces and horizontal and vertical mixing, and then using Cartesian coordinates (Kumar, 2013):

$$
\quad \frac{\partial \overline{v}}{\partial t} + \frac{1}{H}\left[ \frac{\partial}{\partial x}\left( \int uv dz \right) + \overline{v}\left( \frac{\partial}{\partial x}\int u^{st} dz \right) + \overline{v}\left( \frac{\partial}{\partial y}\int v^{st} dz \right) \right] + \tag{1}
$$

$$
\frac{\overline{u^{st}}}{H}\left[ \frac{\partial}{\partial x}\left( \int v dz \right) - \frac{\partial}{\partial y}\left( \int u dz \right) \right] + f\overline{u} + f\overline{u^{st}} = -\frac{1}{\rho_0}\frac{\partial p}{\partial y} + \overline{F^{wy}} + \frac{\tau_s^y}{\rho_0 H} - \frac{\tau_b^y}{\rho_0 H}
$$





where $u$ and $v$ are the along-shore and cross shore velocity components, $u^{st}$ and $v^{st}$ are the Stokes velocities, the overbar indicates depth averaging, $H$ is the total water depth, $f$ is the Coriolis parameter, $\rho_0$ is the reference density, $\tau_s^y$ and $\tau_b^y$ are the surface and bottom stress, respectively, and $F^{wy}$ is the non-conservative wave forcing. Going from left to right, the terms in the equations are local acceleration (*ACC*), horizontal advection (*HA*), horizontal vortex force (*HVF*), Coriolis (*COR*), Stokes–

Coriolis (*StCOR*), pressure gradient (*PG*), non-conservative wave forces (*WF*), surface stress (*SStr*) and bottom stress (*BStr*). Terms in blue are the wave-induced terms.

The pressure gradient term includes (Kumar et al., 2012) the Eulerian non-WEC contribution ($P^c$) and the WEC contribution ($P^{WEC}$), which can be decomposed in a quasi-static response ($P^{qs}$), a Bernoulli head ($P^{bh}$) and a surface pressure boundary correction ($P^{pc}$):

$$\nabla \varphi = P^c + P^{WEC} = P^c + P^{qs} + P^{bh} + P^{pc} \tag{2}$$

The non-conservative wave forcing term $F^{wy}$ includes accelerations due to (Kumar et al., 2012): bottom streaming ($B^{bf}$), surface streaming ($B^{sf}$) and wave breaking ($B^{wb}$). The latter is further decomposed in whitecapping induced acceleration ($B^{wcap}$), bathymetry induced breaking and acceleration ($B^b$) and wave rollers and rollers acceleration ($B^r$):

$$F^{wy} = B^{bf} + B^{sf} + B^{wb} = B^{bf} + B^{sf} + B^{wcap} + B^b + B^r \tag{3}$$

## 2.5 Skill assessment techniques

In order to assess the model behavior, the estimation of bias, the root mean square deviation (RMSD), the Pearson's correlation (Pearson's $r$) and the model skill score ($d$, following the method presented in Willmott (1981)) are undertaken. These values are defined as follows:

$$bias = \frac{1}{N} \sum \left( X_{model} - X_{obs} \right) \tag{4}$$

$$RMSD = \sqrt{\frac{1}{N} \sum \left( X_{model} - X_{obs} \right)^2} \tag{5}$$

$$r = \frac{\sum \left( \left( X_{model} - \overline{X_{model}} \right) \left( X_{obs} - \overline{X_{obs}} \right) \right)}{\sqrt{\sum \left( X_{model} - \overline{X_{model}} \right)^2} \sqrt{\sum \left( X_{obs} - \overline{X_{obs}} \right)^2}} \tag{6}$$

$$d = 1 - \frac{\sum |X_{model} - X_{obs}|^2}{\sum \left( |X_{model} - \overline{X_{obs}}| + |X_{obs} - \overline{X_{obs}}| \right)^2} \tag{7}$$





where $N$ is the number of samples. Pearson's $r$ describes consistent proportional increases or decreases about respective means of the two quantities, but it makes too few distinctions among the type or magnitudes of possible covariations (Willmott, 1981). By contrast, $d$ is not a measure of correlation or association in the formal sense but rather a measure of the degree to which a model's predictions are error-free. Unlike $r$, $d$ is sensitive to differences between the observed and predicted means as

well as to certain changes in proportionality (Willmott, 1981). Note that analogously to $r$, $d$ is measured from 0 to 1, 1 denoting maximum agreement. When computing these metrics, the first 24 h of the model results were rejected, in order to exclude the possible spin-up of the model.

For circular data, e.g. wave direction, the metrics are computed as follows:

$$bias = \tan^{-1}\left(\frac{\frac{1}{N}\sum \sin\left(X_{model} - X_{obs}\right)}{\frac{1}{N}\sum \cos\left(X_{model} - X_{obs}\right)}\right) \tag{8}$$

$$RMSD = \sqrt{-2 \cdot \ln\left(\frac{1}{N}\sum \cos\left(X_{obs} - X_{model}\right)\right)} \tag{9}$$

$$r = \frac{\sum \left(\sin\left(X_{model} - \overline{X_{model}}\right)\sin\left(X_{obs} - \overline{X_{obs}}\right)\right)}{\sqrt{\sum \sin^2\left(X_{model} - \overline{X_{model}}\right)\sum \sin^2\left(X_{obs} - \overline{X_{obs}}\right)}} \tag{10}$$

## 3 Results

### 3.1 Numerical model skill assessment

The ROMS and SWAN models for the same study period and the same model configurations have been validated thoroughly

in previous studies (Ràfols et al., 2017a, b). The aim of this section is to analyze the skill of the coupled run in comparison to the uncoupled runs.

The first step in the numerical skill assessment is to examine the quality of the wind field, which is used to force the numerical models. Table 1 shows the bias, RMSD, $r$ and $d$ obtained from the comparison between the DB measured data and the wind field provided by the SMC. And Fig. 3 presents the time series for the modeled and measured wind intensity at DB. The

comparison shows a slight underestimation of the wind intensity but the main underestimation does not correspond to the NW wind events, which are the focus of this study. During the study period, four NW wind-jet events have been selected (see the red boxes in Fig. 3). These events were previously analyzed in Ràfols et al. (2017b), where statistical metrics for each episode were provided. To see the temporal evolution of a wind jet more clearly, in Fig. 4a the time series during the wind-jet event E3 are presented, which is the event that spans more in space and thus can be observed in the DB location. Overall, the modeled

wind during the wind-jet events is less underestimated and the wind-jet temporal evolutions are properly reproduced.

Table 1 also shows the statistics obtained from the comparison of the measured wave parameters at DB and the modeled ones. The $Hs$ and $Tm_{02}$ time series at DB during the wind-jet event E3 are shown in Fig. 4b and Fig. 4c, respectively. In





**Table 1.** Statistics comparing the wind and the modeled wave parameters with the DB data.

|  |  | bias | RMSD | $r$ | $d$ |
|---|---|---|---|---|---|
| Wind |  | -0.04 m/s | 1.83 m/s | 0.83 | 0.89 |
| $Hs$ | uncS | -0.25 m | 0.38 m | 0.89 | 0.86 |
|  | cRS | -0.28 m | 0.40 m | 0.90 | 0.85 |
| $Tm_{02}$ | uncS | -0.95 s | 1.09 s | 0.79 | 0.67 |
|  | cRS | -0.34 s | 0.52 s | 0.85 | 0.86 |
| $Dir$ | uncS | -9.39° | 34.89° | 0.84 | – |
|  | cRS | -10.34° | 36.37° | 0.84 | – |

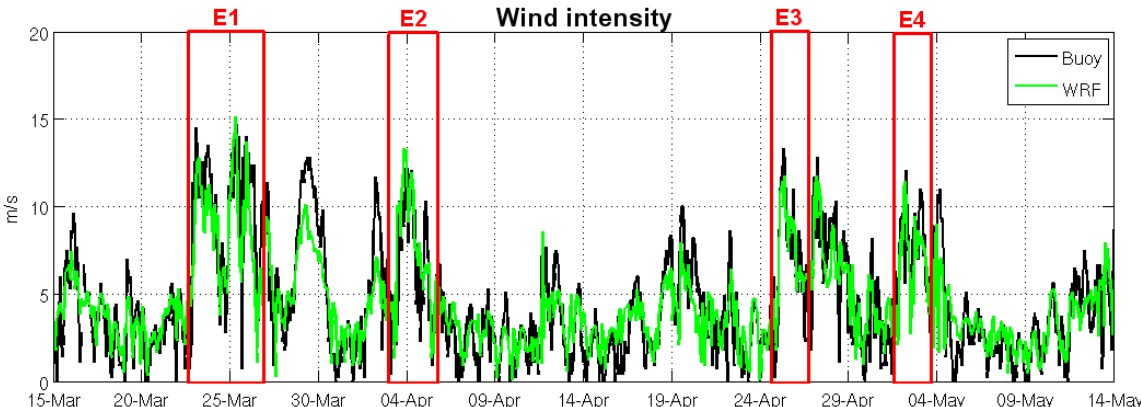

**Figure 3.** Comparison between the wind measured by the DB buoy (black) and the one modeled by the WRF model and used as input for the SWAN and ROMS models (green). See statistics in Table 1. The red boxes are the four wind-jet events.

general, the $Hs$ does not show relevant differences between the *uncS* run and the *cRS* run results. It is important to note the negative bias, which indicates that the $Hs$ parameter is slightly underestimated. This is a clear consequence of the previously mentioned underestimation of the wind. In contrast, $Tm_{02}$ shows a clear improvement when the models are coupled. The mean wave period $Tm_{02}$ is defined as follows:

$$Tm_{02} = 2\pi \left( \frac{\int \int \omega^2 E(\omega,\theta)d\omega d\theta}{\int \int E(\omega,\theta)d\omega d\theta} \right)^{-1/2} \qquad (11)$$

where $E(\omega,\theta)$ is the variance density and $\omega$ is the absolute radian frequency. The latter is determined by the Doppler shift phenomenon with $\omega = \sigma + \mathbf{k} \cdot \mathbf{U}$, where $\sigma$ is the relative radian frequency (i.e. as observed in a frame of reference moving with the current velocity), $\mathbf{k}$ the wave number vector and $\mathbf{U}$ the current vector. In absence of currents, the relative radian frequency equals the absolute radian frequency. It is important to note that the buoy measures at a fixed location (i.e. in an absolute frame) and, for this reason, the comparison of the measured period with the modeled one is more realistic when the results from the



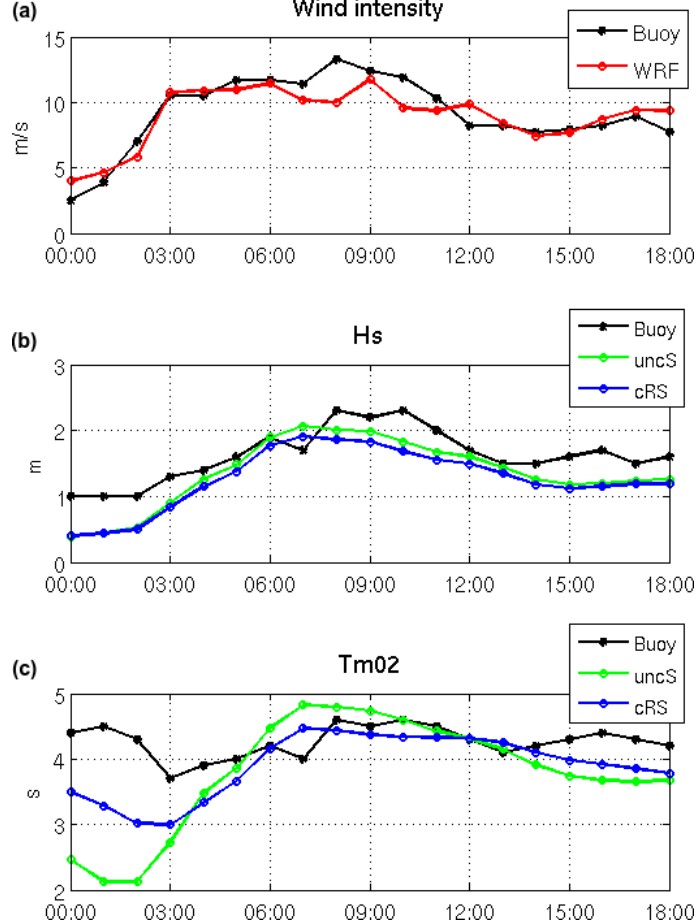

**Figure 4.** (a) Wind intensity, (b) $Hs$ and (c) $Tm_{02}$ time series at DB during the wind-jet event E3 (25 April 2014). In black, the measured data, in red the WRF model data, in green the *uncS* run results and in blue the *cRS* run results.

*cRS* run are used (i.e. the absolute period) instead of the results from the *uncR* run (i.e. the relative period). Therefore, the differences found in the $Tm_{02}$ parameter might be explained, in part but not uniquely, by the differences in frequency due to the Doppler shift phenomena that are included in the wave model when the models are coupled.

Table 2, where the modeled data are compared with measurements from CB, show similar results to Table 1. The most no-
5 ticeable difference between the two tables is the $Dir$ parameter, which shows better agreement in the DB case. The comparison at DB shows very good results with strong correlations and no relevant differences between the *uncS* and *cRS* runs. In contrast, at CB location, the agreement with observations is smaller but a clear improvement of the results is obtained when the currents are considered (i.e. with the *cRS* run).

In Table 3, the modeled water currents are compared with the HF radar surface current measurements. The metrics presented
10 in the table correspond to point P3 and show good agreement, with skill metrics that are in accordance with values found in



**Table 2.** Statistics comparing the modeled wave parameters with the CB data.

|          |      | bias     | RMSD    | $r$  | $d$  |
|----------|------|----------|---------|------|------|
| $Hs$     | uncS | -0.14 m  | 0.23 m  | 0.87 | 0.79 |
|          | cRS  | -0.17 m  | 0.24 m  | 0.89 | 0.79 |
| $Tm_{02}$| uncS | -1.24 s  | 1.43 s  | 0.42 | 0.48 |
|          | cRS  | -0.34 s  | 0.64 s  | 0.71 | 0.79 |
| $Dir$    | uncS | 4.11°    | 33.06°  | 0.46 | –    |
|          | cRS  | -0.99°   | 25.97°  | 0.52 | –    |

previous work when comparing HF radar data with modeled data (Port et al., 2011; O'Donncha et al., 2015; Lorente et al., 2016). Comparing the results from the *uncR* run with the results from the *cRS* run, some differences are observed (e.g. a decrease of the bias is obtained in the cross-shelf velocity component when the models are coupled), but the differences are not relevant enough to discern if one configuration agrees better than the other. A similar conclusion can be reached analyzing

the scatter plots (not shown) comparing the HF radar data with the modeled data at P3. The differences between the *cRS* and *uncR* runs are not relevant, but the modeled cross-shelf components show a better fit with the measurements, with regression slopes of 1.01 for both runs, than the along-shelf components, with regression slopes of 0.64 and 0.68, respectively. In general, the modeled water currents show larger intensities than the measured ones.

**Table 3.** Statistics comparing the modeled water currents at P3 with data from the HF radar.

|    |      | bias        | RMSD        | $r$  | $d$  |
|----|------|-------------|-------------|------|------|
| u  | uncS | -4.20 cm/s  | 14.02 cm/s  | 0.56 | 0.73 |
|    | cRS  | -1.49 cm/s  | 13.71 cm/s  | 0.55 | 0.74 |
| v  | uncS | 3.50 cm/s   | 14.18 cm/s  | 0.65 | 0.77 |
|    | cRS  | 2.88 cm/s   | 14.86 cm/s  | 0.66 | 0.77 |

### 3.2   Description of the wave effects on currents

Fig. 5 compares *uncR* and *cRS* run results during the wind-jet event E3 at P1 (73.7 m depth) and P3 (98.9 m depth) along with HF radar water surface current at P3. The figure also shows the wind intensity evolution at each point and the Hs comparison between the *uncS* and *cRS* run results. The wind-jet event E3 starts on 25 April at 02:00 (UTC), forms very quickly, reaches its maximum intensity at 06:00 and fades gradually. The water current time series show that during the wind jet peak, there is a negative increase of the cross-shelf current component (i.e., offshoreward) and a decrease of the along-shelf current. Then,

after the wind-jet peak, the along-shelf component becomes more negative (i.e., southwestward). Comparing the results from the *uncR* and *cRS* runs, it is observed that larger differences occur at the shallowest point (P1), with differences up to 20 cm/s,





while at P3 both runs present very similar results. No measured data are available for P1, thus it cannot be discerned which run best fits the observation. In contrast, at P3, the modeled results can be compared with the HF radar data but it is difficult to state which simulation best reproduces the observations. The influence of waves at the cross-shelf circulation is limited and the surface circulation of both runs presents similar patterns.

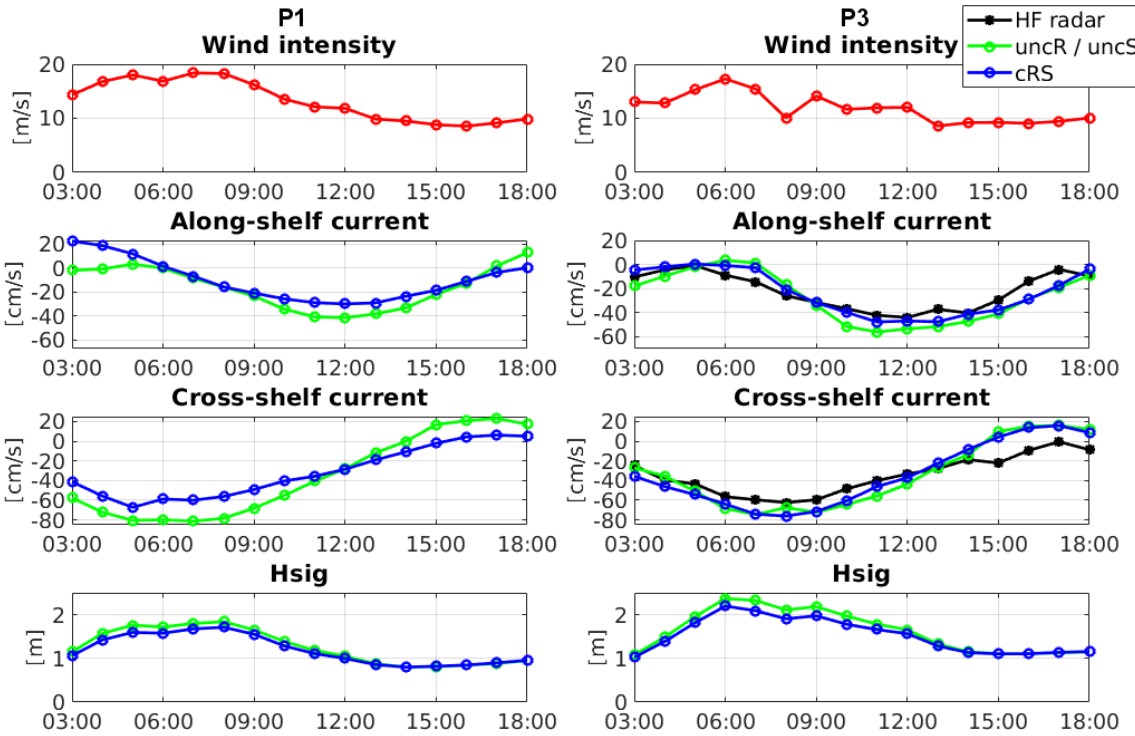

**Figure 5.** Wind intensity, along- and cross-shelf surface currents and Hs time series at P1 and P3 during the wind-jet event E3 (25 April 2014). Negative values mean offshore and southwestwards. In red, the modeled wind intensity, in black the HF radar data, in green the *uncR* and *uncS* runs results and in blue the *cRS* run results.

5  With the aim of visualizing the differences in the current patterns and the spatial variability between the different runs, in Fig. 6 the measured HF radar currents are compared with the surface currents obtained with the *uncR* and *cRS* runs in four snapshots, which correspond to the evolution of the wind-jet event E3. For clarity, the figure presents the results up to the mid-slope. The modeled water currents are more intense than the water currents measured by the HF radar but the circulation patterns are consistent. There are slight differences between the *uncR* run results and the *cRS* run results. An increase of the

10  current intensity is observed at the start of the wind jet when the waves are considered in the ROMS model (Fig. 6c and 6d second column). In addition, the region affected by the wind jet seems to be expanded to the northeast, resulting in stronger water currents in the cRS run. Nevertheless, the main current patterns obtained with both runs are very similar and coincide with the behavior presented in Ràfols et al. (2017a).





**Figure 6.** Results for the wind-jet event E3. (a) 10-m wind intensity; (b) HFR current intensity; (c) *uncR*-modeled surface current intensity; (d) *cRS*-modeled surface current intensity; (e) *cRS*-modeled $Hs$ and mean wave direction. For clarity, the results are shown up to the mid-slope. The CB and DB locations are shown with pink triangles.





Figure 7 shows the $Hs$ and water current mean differences considering: the whole study period, the whole study period except the wind-jet events and just the four wind-jet events. It is observed that the major differences are obtained during the wind-jet events. The mean differences obtained for the whole period are very similar to the mean differences under no wind-jet conditions, with differences shorter than $\pm 5$ cm/s. During wind-jet conditions, a clear decrease of the water current intensity is

5    observed at the wind-jet axis when the waves are considered, but the differences are less than 10 cm/s. In contrast, at shallow regions, the water current intensities are increased, showing differences up to $15-20$ cm/s. An increase of the current intensity is also observed at the northeast corner of the domain but there the differences are just around 5 cm/s.

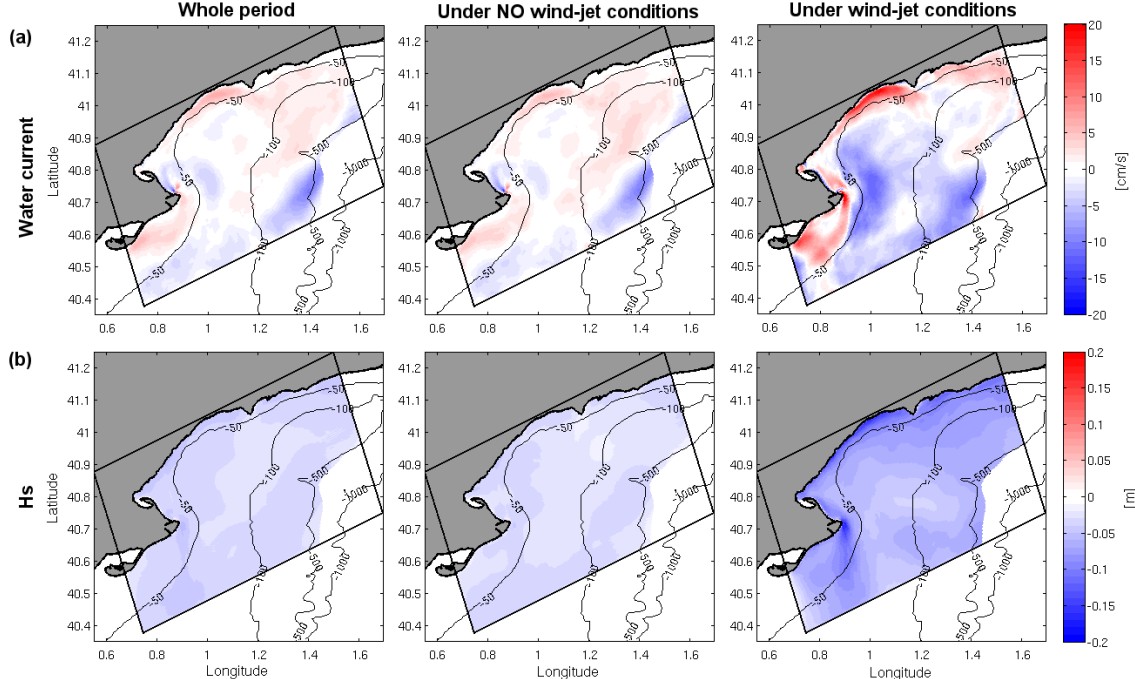

**Figure 7.** (a) Water current and (b) $Hs$ mean differences between the *uncS/uncR* run and the *cRS* run considering the whole period (left), the whole period except the wind-jet events (centre) and during the four wind-jet events (right). Positive values correspond to *cRS* value $>$ *uncS/uncR* value.

The evolution of the buoyancy or Brunt–Väisälä frequency ($N = \sqrt{-\frac{g}{\rho_0}\frac{\partial \rho(z)}{\partial z}}$, where $\rho_0$ is the reference density and $g$ is the gravitational acceleration) is investigated in order to analyze the differences between the *uncR* and the *cRS* run results in the

10    water column structure. Figure 8 shows the Brunt–Väisälä frequency evolution (before and after the wind jet) at P3 during the four wind-jet events for both *uncR* and *cRS* runs. It is observed that the vertical structure of the water column is significantly different when the waves are taken into account. The *cRS* run always presents a less stratified water column, both before and after the wind jet. When a wind jet occurs, the expected behavior is that the water column will become less stratified after the wind jet than before it. This is observed in all the studied wind-jet events but the surface mixed-layer depth (SML; i.e. the




distance from the surface until the top of the pycnocline) after the wind jet obtained with the *cRS* run is larger (i.e. deeper) than the one obtained by the *uncR* run. Thus, the vertical mixing is significantly enhanced when the waves are taken into account.

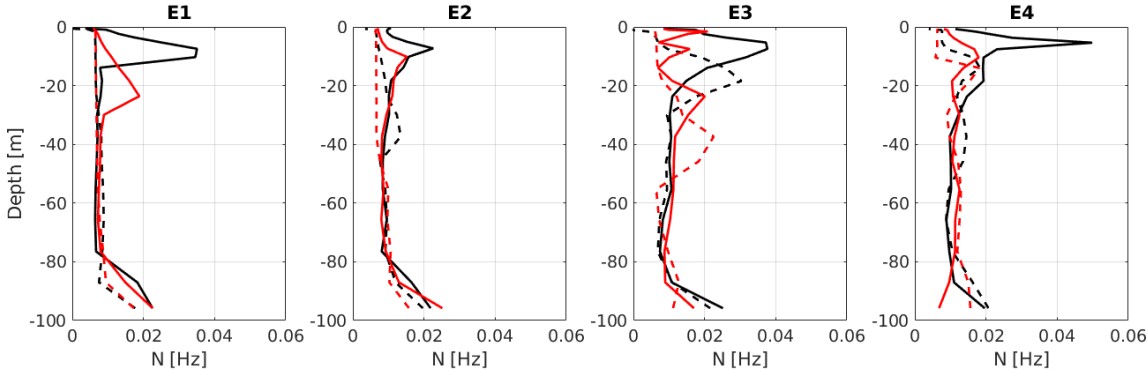

**Figure 8.** Comparison of the Brunt–Väisälä frequency at the start (solid line) and the end (dashed line) of each wind-jet event obtained from the results of the *uncR* (in black) and *cRS* run (in red) at P3.

Analyzing the results from *uncR* and *cRS* runs, it is found that there is a clear enhancement of the TKE when the waves are considered, also with some increase of the SStr (see Fig. 9). Note that the SStr felt by the ocean is equal to the air-side stress,

5    which in the cRS run include the wave-dependent sea surface roughness, but it does not account for the stress acting on waves and the dissipation due to wave breaking. The mean TKE and SStr values obtained with the model during the wind-jet event E3 at P3 shift from $8.14{\times}10^{-4}$ m$^2$/s$^2$ and 0.25 N/m$^2$ with the *uncR* run to $5.13{\times}10^{-3}$ m$^2$/s$^2$ and 0.31 N/m$^2$ with the *cRS* run. Additionally, the TKE and SStr peak coincide with the wind jet peak (25 April at 06:00) and the peak values found at P3 are $2.44{\times}10^{-3}$ m$^2$/s$^2$ and 0.75 N/m$^2$ for the *uncR* run and $1.11{\times}10^{-2}$ m$^2$/s$^2$ and 0.88 N/m$^2$ for the *cRS* run. Thus, the TKE is 1

10    order of magnitude stronger when the waves are considered, which leads to an enhancement of the water column mixing and thus a decrease of the stratification.

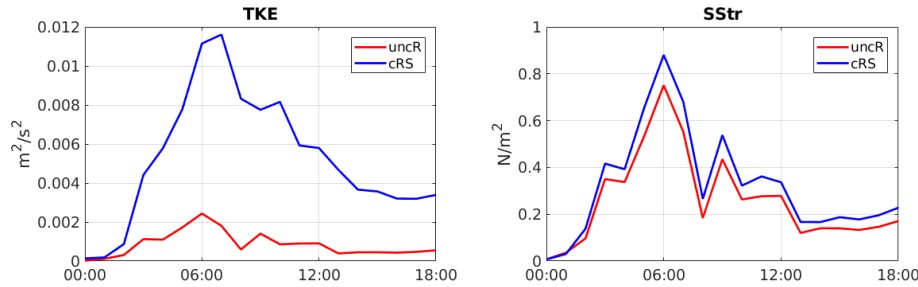

**Figure 9.** Time series comparison of the TKE (left) and SStr (right) obtained from the results of the *uncR* (in red) and *cRS* run (in blue) at P3 during the wind-jet event E3 (25 April 2014).

In order to evaluate how the waves' effects are taken into account in the momentum balance, the terms of equation 1 are analyzed. During a calm period before the wind jet, the cross-shelf momentum balance is between the *PG* and *COR* terms, and



the remaining terms are (at least) 1 order of magnitude smaller. Thus, the wave effects on the momentum balance are negligible. In contrast, during a wind-jet event, more terms are involved in the cross-shelf momentum balance. From the coastline until 4 km offshore ($\sim$50 m depth), the *WF* term ($1.85\times10^{-5}$ m/s$^2$) is on the same order of magnitude as the *PG* ($2.42\text{x}10^{-5}$ m/s$^2$), *SStr* ($2.73\times10^{-5}$ m/s$^2$) and *HA* ($1.99\times10^{-5}$ m/s$^2$) terms. From that point until tens of kilometers offshore the *PG*

($1.34\times10^{-5}$ m/s$^2$) term is mainly balanced by the *SStr* ($1.07\times10^{-5}$ m/s$^2$) and *WF* ($5.05\times10^{-6}$ m/s$^2$) terms, also including some contribution from the *COR* and *HA* terms. However, the *WF* term weight is half the weight of the *SStr* term. Thus, the *WF* term included by the VF formalism plays an important role in the momentum balance in the first kilometers offshore (i.e. in coastal regions). Analyzing the *WF* term, it is found that its main contributor is the surface streaming ($B^{sf}$; $1.65\times10^{-5}$ m/s$^2$ and $3.94\times10^{-6}$ m/s$^2$ for shallow and deep water, respectively), especially in shallow waters, with also some contribution of

the wave breaking term ($B^b$; $2.01\times10^{-6}$ m/s$^2$ and $1.11\times10^{-6}$ m/s$^2$,respectively). Regarding the *PG* term, its weight is mainly due to the non-WEC contributions ($P^c$; $1.60\times10^{-5}$ m/s$^2$ and $1.37\times10^{-5}$ m/s$^2$, respectively) together with some contribution of the quasi-static response ($P^{qs}$; $1.37\times10^{-5}$ m/s$^2$ and $3.21\times10^{-6}$ m/s$^2$).

## 3.3   Description of the current effects on waves

The irregular nature of wind causes irregular wind waves of different heights, periods and directions. For this reason, wind

waves are usually described using spectral techniques, where the random motion of the sea surface is treated as a summation of harmonic wave components. In Fig. 10 the wave response during a wind-jet event is analyzed in terms of the variance density spectrum $E(f, \theta)$ evolution obtained from the numerical model. The one- and two-dimensional frequency–direction spectra evolution at P2 (i.e. at the wind-jet axis) obtained with the *uncS* and *cRS* runs during the wind-jet event E3 are compared. The runs show similar spectra evolution patterns. When the wind jet starts, the wave field is adapted to the new wind, generating a

bimodal spectrum with a wider peak at the NW (which is consistent with the new wind direction, i.e., it is a new sea system) and a peak at the south corresponding to the "old" sea system. At the peak of the wind jet, the spectra are dominated by the new sea system and, when the wind-jet intensity diminishes, another new sea system occurs, while the energy due to the wind jet decrease gradually. In addition, a swell system appears at the northeast, due to the coexistence of NW wind at the region and northerly wind at the northern part of the coast (Ràfols et al., 2017b). The main difference between the *uncS* and *cRS*

runs is that the spectra obtained with the *cRS* run present less energy at the peak than the *uncS* run. An energy increase at higher frequencies (i.e. at the spectrum tail) can also be observed when the currents are considered, but overall the *uncS* run presents more energy. A less energetic spectrum means shorter $Hs$ values, which is consistent with the values obtained from the numerical results.

    In Fig. 5 and Fig. 6, it is observed that, during the wind-jet event, the wave field responds directly to the wind. In Fig. 6,

the 2D $Hs$ maps show a clear increase of the wave height at the wind-jet axis that, at the wind-jet peak, reaches values up to 2.43 m. The time series presented in Fig. 4 and Fig. 5 show that the $Hs$ diminish when the water currents are considered and that the major differences ($\sim$15–20 cm) occur during the wind-jet peak. Similar results are shown in Fig. 7, where the mean differences show that the $Hs$ from the *cRS* run tend to be shorter than the $Hs$ from the *uncS* run and that the major differences are observed during the wind-jet events. Under such conditions, the mean differences at shallow regions reach values of 15





**Figure 10.** Spectra evolution during event E3 at P2. (a) 2D spectrum from *uncS* run. (b) 2D spectrum from *cRS* run. (c) 1D spectra from both runs and the corresponding *Hs* values. The arrows shown in a) and b) indicate the direction and magnitude of wind (red) and current (blue).

cm, while the mean difference at the wind-jet axis is around 6 cm. Comparing the results from the *cRS* and the *uncS* runs, it is

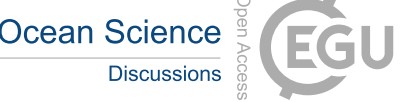



found that considering the water currents produces a mean effect of 11% in the *Hs* parameter at CB location and a mean effect of 4% at DB location.

In order to analyze the *Hs* differences obtained with the two runs, Fig. 11 shows the differences in *Hs* at P2, distinguishing the differences in the wave and current propagation directions. It is found that the *Hs* from the *cRS* run tends to be shorter

5  (stronger) than the *Hs* from the *uncS* run when the difference between the propagation direction of waves and currents is shorter (stronger). This is to say that including the current effects on waves results in a decrease (increase) of *Hs* when the waves and the currents propagate in the same (opposite) direction. In general, the *Hs* differences between the runs are small ($\Delta Hs < 5$ cm). However, during the NW wind jets these differences increase up to 10–14 cm and, in the case of event E3, reach 20 cm. The mean differences observed at this point correspond to 10% of *Hs*.

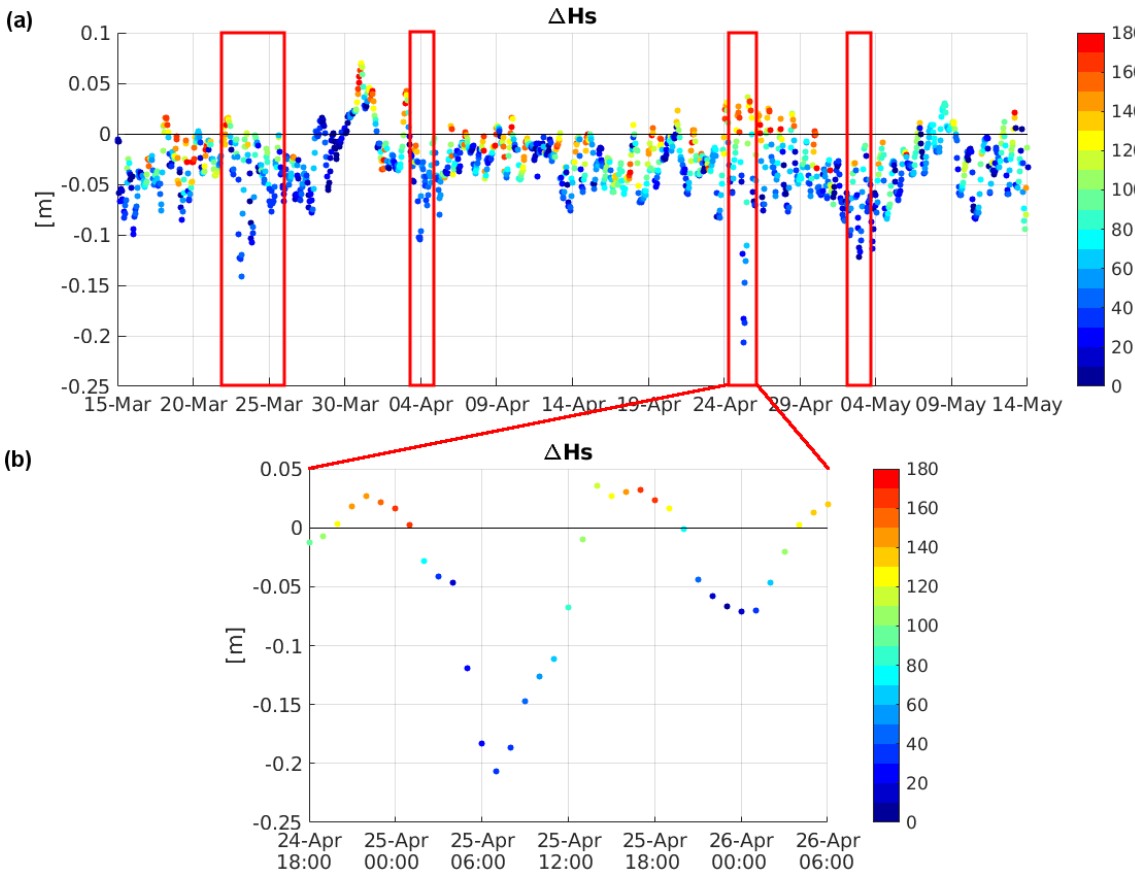

**Figure 11.** (a) *Hs* differences at P2. The different colors correspond to the angle between the directions of wave and current propagation. (b) Detailed view of the period corresponding to event E3.

10  The $Tm_{02}$ obtained with the *cRS* tends to be longer than the one obtained with the *uncS* excepting the wind-jet event periods, where the $Tm_{02}$ from the coupled run is shorter (see Fig. 4). This is consistent with the frequency increase in the *cRS* run detected in the spectra analysis during the wind-jet event E3. Figure 12 shows the $Tm_{02}$ and *Dir* time series obtained



with the *uncS* and the *cRS* runs compared to the CB measured data. Note that the CB location is not affected by the wind jet. Qualitatively, the *cRS* run shows a clear improvement in the agreement of the $Tm_{02}$ results with the measurements, which is consistent with the statistical parameters collected in Tables 1 and 2. Comparing the results from the *cRS* and the *uncS* runs, it is found that considering the water currents produces an average effect of 48% in the $Tm_{02}$ parameter at the CB location. This

5 effect is reduced to 27% at the DB location.

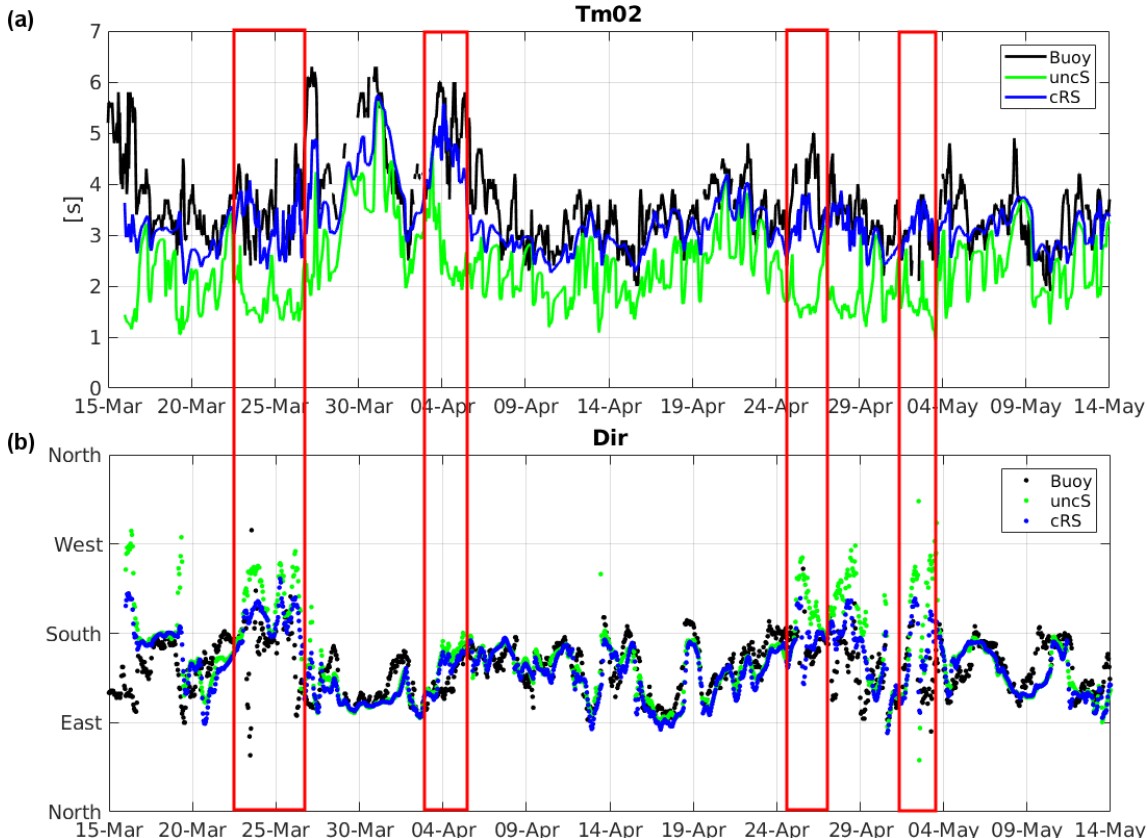

**Figure 12.** Comparison of the $Tm_{02}$ and $Dir$ parameters time series obtained with the *uncS* (green) and *cRS* (blue) runs with the data measured by CB (black). Note that the first 24 h of the model results have been rejected.

Regarding the mean wave direction, no relevant differences are observed between the *uncS* and *cRS* runs at deep water (not shown). However, similarly to the results presented in Table 2, Fig. 12b shows that at the CB location (i.e. in shallow waters) the mean wave direction is improved during the wind-jet events. Analyzing the mean wave direction differences between the *uncS* and *cRS* runs, it is found that relevant differences occur near the coastline.



## 4    Discussion

### 4.1    Effects of waves on the current field

The main differences between the *uncR* and *cRS* runs have been detected in the water column structure. The vertical mixing of the water column is stronger when the waves are considered. This behavior can be explained by the TKE injection and the use of a wave-dependent sea surface stress in the *cRS* run. Similar results have been observed in previous work. Rong et al. (2014) studied the WCI over the Texas–Louisiana Shelf and found that the wave effects can redistribute the freshwater both vertically and horizontally and thus affect the stratification. Bruneau and Toumi (2016) also found that the mixed-layer depths were enhanced in presence of waves. Niu and Xia (2017) investigated how the Lake Erie dynamics were impacted by the wave-induced surface stress and found that it produced an enhancement of the surface mixing and a weakening of the stratification strength. It is important to note that, although the results presented in this study are consistent, there are no available measurements to verify them. Thus, it can not be stated if the *cRS* run is more adjusted to the reality or if it is "over-mixing".

The results presented above show that including the wave effects does not produce a relevant difference to the water current velocity during a wind-jet event and has a weak impact on the water circulation patterns. Similar results were presented by Bruneau and Toumi (2016), who analyzed the wave-induced processes at the Caspian Sea and found that they have a weak impact on the dynamics of the region. The momentum balance analysis has shown that the WF term is one of the leading terms in very shallow areas (until ∼50 m water depth). For this reason, using a numerical domain at a more coastal scale with water depths up to 50 m would probably show more effects at the current field, rather than the domain used in this work, which is focused on the inner shelf, where the water depth reaches values higher than 100 m. As a matter of fact, Osuna and Wolf (2005) studied the WCI in the Irish Sea and found that the effect of waves on currents are evident in the eastern coastal areas, with daily mean current differences larger than 10 cm/s during strong wave events.

### 4.2    Current effects on the wave field

The numerical results present an improvement in the $Tm_{02}$ parameter when the coupling effects were considered (see Fig. 12a). Consequently, the inclusion of the current velocity in the estimation of wave period is not negligible, and it must be considered if high-quality modeling is required (similar to Bolaños et al. (2014)). It should be noted that the results show that the effects of currents on the wave field are stronger for the $Tm_{02}$ parameter than for the *Hs* parameter. For instance, Osuna and Monbaliu (2004) found that the effect of coupling is 1 order of magnitude stronger for the $Tm_{02}$ parameter (about 20%) than in the case of *Hs* (about 3%).

During a wind-jet event, a decrease of the *Hs* is found when the currents are taken into account. The decrease (increase) of *Hs* in the presence of an opposite (following) current is a well-known effect that has been investigated before by several authors (e.g. Benetazzo et al., 2013; Dutour Sikirić et al., 2013; Viitak et al., 2016). For example, Benetazzo et al. (2013) studied the WCI at the semi-enclosed Gulf of Venice and found that during Bora conditions, with the currents propagating in the same





direction as waves, the comparison between coupled and uncoupled models showed a reduction of *Hs* on the order of 0.6 m when the waves were considered.

The differences in mean wave direction found in shallow waters could be due to the current-induced refraction (Wolf and Prandle, 1999; Olabarrieta et al., 2011). However, it is important to note that these differences were found very near the

coastline, specifically until 2 km offshore. Since the model mesh resolution is of 350 m, there are very few grid points and thus it is not possible to extract a concise conclusion about this phenomenon with the results obtained in this study. A study at more coastal scales would be necessary to discern such processes.

Finally, considering the currents causes wave spectral reshaping. During a cross-shelf wind-jet event, the presence of currents induces a shoaling-like process. In general, a reduction of the energy peak and a slight increase of the energy at the tail of the

spectrum is observed. This is consistent with the results presented in Fan et al. (2009), where the authors found that when the wave–current interactions were considered, the peak of the frequency spectrum was reduced and shifted toward higher frequency. Rusu (2010) also found that the presence of currents leads to a redistribution of the wave energy over the spectrum.

## 5  Conclusions and future works

The wave–current interactions have been investigated using numerical models. Three different runs have been performed: an

uncoupled ROMS run, an uncoupled SWAN run and a two-way coupled run. The comparison among these runs shows that at the continental shelf the surface water current presents similar results in the coupled and the uncoupled configurations and the momentum balance analysis reveals that the non-conservative wave forcing (WF) term plays an important role in shallow waters. The results show that including wave effects induces major mixing of the water column (the SML increase), mainly due to the TKE injection and the enhanced surface stress. Additionally, when the water currents are considered in the waves

forecast, wave spectral reshaping occurs, the $Tm_{02}$ improves and the wave energy (and thus the *Hs*) diminishes (increases) when the water currents and waves propagate in the same (opposite) direction. The results also indicate that more processes occur in shallower waters, e.g. current-induced refraction, but a more coastal domain with a finer grid is necessary to evaluate them.

Overall, the numerical results have demonstrated to be physically reasonable, being capable of reproducing the well known

coupling effects. This has allowed to investigate the impact of the WCIs but more measurements would be needed in order to perform a more quantitative analysis. Thus, in the future it would be interesting to perform some measurement campaigns to enable more accurate model validation and more exhaustive analysis of the dynamics of the region. In addition, it would be interesting to investigate the role of the sea surface roughness coupling the ROMS and SWAN models with the WRF model.

*Data availability.* HF radar data and buoy measurements used in this contribution can be consulted in http://portus.puertos.es, the IBI-

MFC model data is available in http://marine.copernicus.eu and the WRF model data was provided by the Catalan Service of Meteorology (http://meteo.cat/). Data processing and displaying was done using a licensed Matlab program.



## Appendix A: The logarithmic wind profile

The logarithmic wind profile used to extrapolate from 10 m to 3 m the modeled wind is as follows

$$U_z = \frac{U^*}{\kappa} \ln\left(\frac{z}{z_0}\right) \tag{A1}$$

where $U_z$ is the mean horizontal wind velocity at a given height $z$, $U^*$ is the frictional velocity, $\kappa$ is the von Kármán constant

($\simeq 0.4$) and $z_0$ is the aerodynamic roughness length.

The roughness length is estimated by means of the Charnock's relation

$$z_0 = \frac{\alpha_{CH} U^{*2}}{g} \tag{A2}$$

where $g$ is the gravitational constant and $\alpha_{CH}$ is the Charnock parameter (in this study it has been considered an $\alpha_{CH}$ equal

to 0.011).

The friction velocity is related to the known wind speed at 10 m elevation ($U_{10}$) with

$$U^{*2} = C_D U_{10}^2 \tag{A3}$$

where $C_D$ is the drag coefficient from Wu (1982)

$$C_D(U_{10}) = \begin{cases} 1.2875 \cdot 10^{-3}, & \text{for } U_{10} < 7.5 \text{ m/s} \\ (0.8 + 0.065 \cdot U_{10}) \cdot 10^{-3}, & \text{for } U_{10} \geq 7.5 \text{ m/s} \end{cases} \tag{A4}$$

*Competing interests.* The authors declare that they have no conflict of interest.

*Acknowledgements.* The development of this research was partially funded by the Doctorats Industrials 2013 PhD program of the Catalan Government. The authors also acknowledge Puertos del Estado for the data set provided. This work received funding from the EU H2020 program under grant agreement no. 730030 (CEASELESS project).

©c Author(s) 2018. CC BY 4.0 License.

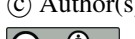



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
