# Peer review of "Wave-Current Interactions in a Wind-jet Region"

_Ocean Science, 2018_

## Referee Comment (RC1) · Anonymous Referee #1 · 25 Oct 2018

This paper is well written and an interesting contribution to the field. I particularly enjoyed the analysis of TKE and the Brunt-Väisälä frequency, showing that the vertical mixing of the water column is stronger when waves are considered. I therefore recommend it to be published after the following issues are addressed.

Do you think there is any tide included in the results shown in Fig. 5? There appears to be a roughly 12.5 hour period to the oscillation, and this could be dominating the time series and masking the effect of the wind-jet. I know the tide is probably very small in this region, but it should be considered. I suggest you perform a harmonic analysis on the time series (make sure it is long enough) and subtract the tide from the time series, so you are left with a non-tidal residual. If the small changes during the wind-jet you describe are due to the wind-jet, then they should still be there.

[Figure]

Page 6 Lines 4 – 9: Please state what boundary forcing was imposed on the nested models, i.e. water elevations and/or currents (barotropic / baroclinic), temperature, salinity?

The naming of the model runs uncS, cRS and uncR are not well defined. They first appear in Table 1 and are used in a number of the figures. I suggest you define them in the text in the first paragraph in section 3.1. On reading the text it becomes evident what they are/mean but it is confusing at first.

In section 3.2 (page 12, line 12) the wind-jet event is said to start at 02:00 UTC, yet Fig. 5 only starts at 03:00. I suggest you start the x-axis at 02:00 to correspond to the text. Also, please label the x-axis in Fig. 5, Time (UTC) or similar.

Minor Comments / Revisions:

Page 1 Line 9: leading a larger mixed-layer depth –> leading to a larger mixed-layer depth

Page 2 Line 29: has been previously –> has previously been

Page 2 Lines 30-31 in the study region and the wind-wave characterization, and water shelf circulation was investigated –> in the study region, and the wind-wave characterization and water shelf circulation were investigated

Page 3 Line 22: data obtained–> data were obtained

Page 3 Line 22: and an high-frequency –> and high-frequency

Page 3 Line 33: KHz –> kHz

Page 4 Line 8: and a two-ways coupling run –> and a model run with two-way coupling

Page 4 Line 11: is N needed?

Page 9 Line 19: And Fig. 3 –> Fig. 3

Page 12 Line 14: What do you mean by "negative increase" exactly? Does that make

it a decrease, or do you mean it becomes more negative. You could say that the magnitude of the current increases. Please rephrase.

Fig. 5: add x-axis label, Time (UTC) or similar.

Page 14 Line 1: What do you mean by "water current" exactly? Do you mean depth-mean?

Page 14 Line 1: what do you mean by "mean differences"? I think you mean the "mean of the hourly instantaneous difference". I would rephrase.

Fig. 9: I suggest you use the same x-axis range as Fig. 5, or would this make it harder to make your point in the text?

Page 22 Line 25: . . . results have demonstrated to be physically reasonable, being capable of reproducing the well . . . –> . . . results are physically reasonable, as they reproduce the well . . .

Page 22 Line 25: This has allowed to investigate the impact of the WCIs . . . –> The results have enabled the WCIs to be investigated . . .

---

## Referee Comment (RC2) · Anonymous Referee #2 · 31 Oct 2018

This work presents some results provided by a ocean/wave high resolution coupled model, comparing with uncoupled runs and observations.

I would suggests to clarify the conclusions in the abstract. For example, it is said that 'the agreement of the modeled wave period improves...', but not respect to what.

I would like to see in the introduction how previous research work relates to the current research. For example, given that this work uses a high resolution model (350m), if the coupling influence depends on resolution in some way.

Very often the authors comment on 'the current effect on waves', and care should be taken here as they are also coupling the sea surface height and the effect of both will have an influence in the results. Furthermore, in a two-way coupled model there will

be a feedback between one model and the other, so that what they will observe will be the overall effect of coupling one model to the other.

The text should clarify if the instantaneous values of the coupling fields are passed between models at every coupling time step (20 minutes), or the average value between coupling steps.

Table 1 should clarify if the winds are the 10m winds or the winds interpolated to 3m. In the text or the table caption it is not well described the meaning of 'uncS' or 'cRS'. In the text some expressions such as Tm02 are used before their meaning is explained.

The surface stresses are calculated by the changes in surface roughness. The expression for the surface roughness here is different to the one used to interpolate 10m to 3m winds, and it should be clarified why the same expression is not used in both cases. In the second case, there is the possibility of using the actual Charnock parameter that can be provided by the wave model, instead of using a default value.

It should be better justified why it is considered that 24 hours are enough to spin-up the model.

The decision of using 24 h is based on different things: our knowledge of our models behavior, the analysis of the time series and the model configurations. We have to keep in mind that the ROMS model is initialized with data from IBI-MFC, so the spin-up time is expected to be short. A brief explanation has been added in the manuscript (page 9 lines 15-17).

One important conclusion is that the largest differences between coupled and uncoupled runs take place at shallower areas, but this is illustrated just by comparing results in two points in the domain. What I miss is a whole domain picture showing differences in some variable between coupled and uncoupled results to actually confirm that the largest differences occur at shallow places, instead of resulting of a fortunate selection of comparison sites.

The article is centered in wave effects on currents, but might be it would be useful to look at other variables such as sea temperature or salinity, as they might better illustrate the effect of vertical mixing.

---

## Author Comment (AC1) · 31 Oct 2018

The authors acknowledge the helpful comments and corrections of Referee #1, which helped to improve the quality of the manuscript. Below, each comment is answered point-by-point. A marked-up version of the manuscript with the corrections is enclosed as a supplement file.

This paper is well written and an interesting contribution to the field. I particularly enjoyed the analysis of TKE and the Brunt-Väisälä frequency, showing that the vertical mixing of the water column is stronger when waves are considered. I therefore recommend it to be published after the following issues are addressed.

[Figure]

Do you think there is any tide included in the results shown in Fig. 5? There appears to be a roughly 12.5 hour period to the oscillation, and this could be dominating the time series and masking the effect of the wind-jet. I know the tide is probably very small in this region, but it should be considered. I suggest you perform a harmonic analysis on the time series (make sure it is long enough) and subtract the tide from the time series, so you are left with a non-tidal residual. If the small changes during the wind-jet you describe are due to the wind-jet, then they should still be there.

Right, the current time series were not filtered and included the tide. We have subtracted the subinertial time series using the same filter we used in a previous study (Ràfols et al., 2017a). Figure 5 has been changed and the filter explanation included in the new version of the manuscript.

Page 6 Lines 4 – 9: Please state what boundary forcing was imposed on the nested models, i.e. water elevations and/or currents (barotropic / baroclinic), temperature, salinity?

The nesting between each SWAN domain consist on providing the energy spectra from the coarser domain to the boundary of the smaller domain. We have added this explanation in the manuscript. The boundary forcing of the ROMS model is the one that includes water elevations, currents, temperature and salinity, but this is explained in Page 6 Lines 21-29 of the revised version.

The naming of the model runs uncS, cRS and uncR are not well defined. They first appear in Table 1 and are used in a number of the figures. I suggest you define them in the text in the first paragraph in section 3.1. On reading the text it becomes evident what they are/mean but it is confusing at first.
We have added the naming of the model runs at the first paragraph in section 2.3.2, where the system set-up is explained and the three model runs are first mentioned.

In section 3.2 (page 12, line 12) the wind-jet event is said to start at 02:00 UTC, yet Fig. 5 only starts at 03:00. I suggest you start the x-axis at 02:00 to correspond to the text. Also, please label the x-axis in Fig. 5, Time (UTC) or similar.

Right, it has been corrected. Now the Figure starts at 02:00 UTC and the x-axis label has been added.

Minor Comments / Revisions:

Page 1 Line 9: leading a larger mixed-layer depth → leading to a larger mixed-layer depth

Corrected.

Page 2 Line 29: has been previously → has previously been

Corrected.

Page 2 Lines 30-31 in the study region and the wind-wave characterization, and water shelf circulation was investigated → in the study region, and the wind-wave characterization and water shelf circulation were investigated

Corrected.

Page 3 Line 22: data obtained → data were obtained

Corrected.

Page 3 Line 22: and an high-frequency → and high-frequency

Corrected.

Page 3 Line 33: KHz → kHz

Corrected.

Page 4 Line 8: and a two-ways coupling run → and a model run with two-way coupling

Corrected.

Page 4 Line 11: is N needed?

No, it has been deleted.

Page 9 Line 19: And Fig. 3 → Fig. 3

Corrected.

Page 12 Line 14: What do you mean by "negative increase" exactly? Does that make it a decrease, or do you mean it becomes more negative. You could say that the magnitude of the current increases. Please rephrase.

Right, this explanation was not clear. It has been rephrased.

Fig. 5: add x-axis label, Time (UTC) or similar.

Done.

Page 14 Line 1: What do you mean by "water current" exactly? Do you mean depth-mean?

No, it is the surface current. It has been changed.

Page 14 Line 1: what do you mean by "mean differences"? I think you mean the "mean of the hourly instantaneous difference". I would rephrase.

Right, it has been rephrased.

Fig. 9: I suggest you use the same x-axis range as Fig. 5, or would this make it harder to make your point in the text?
Yes, it is reasonable to use the same x-axis range. It has been corrected.

Page 22 Line 25: . . . results have demonstrated to be physically reasonable, being capable of reproducing the well . . . → . . . results are physically reasonable, as they reproduce the well . . .

Corrected.

Page 22 Line 25: This has allowed to investigate the impact of the WCIs . . . → The results have enabled the WCIs to be investigated . . .

Corrected.

**Supplement:**

**Wave–Current Interactions in a Wind-jet Region**

Laura Ràfols[a,b], Manel Grifoll[a], and Manuel Espino[a]

[a]Maritime Engineering Laboratory (LIM-UPC), Polytechnic University of Catalonia (BarcelonaTech), C/ Jordi Girona 1-3 Edif. D1, 08034 Barcelona, Spain
[b]Meteorological Service of Catalonia (SMC), C/ Berlin 38-48 4a, 08029 Barcelona, Spain

**Correspondence:** Laura Ràfols (laura.rafols@upc.edu)

**Abstract.** Wave–Current Interactions (WCIs) are investigated. The study area is located at the northern margin of the Ebro Shelf (northwestern Mediterranean Sea), where episodes of strong cross-shelf wind (wind jets) occur. The aim of this study is to validate the implemented coupled system and investigate the impact of WCIs on the hydrodynamics of a wind-jet region. The Coupled Ocean–Atmosphere–Wave–Sediment Transport (COAWST) modeling system, which use Regional Ocean Model System (ROMS) and Simulating WAves Nearshore (SWAN) models, is used in a high-resolution domain (350 m). Results from uncoupled numerical models are compared with a two-way coupling simulation. The results do not show substantial differences in the water current field between the coupled and the uncoupled runs. The main effect observed when the waves are considered is in the water column stratification, due to the turbulent kinetic energy injection and the enhanced surface stress, leading to a larger mixed-layer depth. Additionally, when the water currents are considered, the agreement of the modeled wave period significantly improves and the wave energy (and thus the significant wave height) decreases when the current flows in the same direction as the waves propagate.

*Copyright statement.* The works published in this journal are distributed under the Creative Commons Attribution 4.0 License. This licence does not affect the Crown copyright work, which is re-usable under the Open Government Licence (OGL). The Creative Commons Attribution 4.0 License and the OGL are interoperable and do not conflict with, reduce or limit each other.

[revised manuscript text omitted]

---

## Author Comment (AC2) · 15 Nov 2018

The authors acknowledge the helpful comments and corrections of Referee 2, which helped to improve the quality of the manuscript. Below, each comment is answered point-by-point. A marked-up version of the manuscript with the corrections is enclosed as a supplement file. This version also include the corrections due to the comments by Referee #1.

[Figure]

This work presents some results provided by a ocean/wave high resolution coupled model, comparing with uncoupled runs and observations.

I would suggests to clarify the conclusions in the abstract. For example, it is said that 'the agreement of the modeled wave period improves...', but not respect to what.

The results explanation in the abstract has been improved.

I would like to see in the introduction how previous research work relates to the current research. For example, given that this work uses a high resolution model (350m), if the coupling influence depends on resolution in some way.

A new paragraph has been added at the introduction section in order to relate with previous work about WCIs. A comment on the grid resolution dependency has also been included.
Very often the authors comment on 'the current effect on waves', and care should be taken here as they are also coupling the sea surface height and the effect of both will have an influence in the results. Furthermore, in a two-way coupled model there will be a feedback between one model and the other, so that what they will observe will be the overall effect of coupling one model to the other.

Right, with "the current effect on waves" we wanted to say the effect on the wave field when the models were coupled, i.e. when the wave model included the effects of being coupled with the circulation model but not only and strictly the "current effects". This expression has been changed by "coupling effects on waves" all along the manuscript.

Besides, in order to follow the same criteria, the expression "wave effects on currents" has been changed to "coupling effects on currents".

The text should clarify if the instantaneous values of the coupling fields are passed between models at every coupling time step (20 minutes), or the average value between coupling steps.

The coupling uses instantaneous fields. The explanation in the manuscript has been improved (page 7 lines 14-15).

Table 1 should clarify if the winds are the 10m winds or the winds interpolated to 3m.

In order to be able to compare the modeled winds with the measured ones, the winds in Table 1 are at 3m. It has been specified in the table caption and in the manuscript text.

In the text or the table caption it is not well described the meaning of 'uncS' or 'cRS'.

Due to the previous revision of Referee #1, the naming of the different runs were included in the manuscript text in the first paragraph of section 2.3.2.

In the text some expressions such as $Tm_{02}$ are used before their meaning is explained.

Right, the $Tm_{02}$ was used before it was defined. This has been corrected in the new version of the manuscript.

The surface stresses are calculated by the changes in surface roughness. The expression for the surface roughness here is different to the one used to interpolate 10m to 3m winds, and it should be clarified why the same expression is not used in both cases. In the second case, there is the possibility of using the actual Charnock parameter that can be provided by the wave model, instead of using a default value.

As we understand it, two methods have to be distinguished. On the one hand, there is the formula used to extrapolate the wind data in order to be able to compare them with the measurements. This is used to calculate the statistical parameters (i.e., analyze the wind data quality) and to find the wind-jet events. On the other hand, there are the formulas used by the numerical model to compute the surface roughness, which are different in the uncR run and the cRS run (in the second case it will depend on the wave parameters but in the first case it will not). Maybe this could lead to some confusion, but we think it is important not to merge these different methodologies.

It should be better justified why it is considered that 24 hours are enough to spin-up the model.

The decision of using 24 h is based on different things: our knowledge of our models behavior, the analysis of the time series and the model configurations. We have to keep in mind that the ROMS model is initialized with data from IBI-MFC, so the spin-up time is expected to be short. A brief explanation has been added in the manuscript (page 9 lines 15-17).

One important conclusion is that the largest differences between coupled and uncoupled runs take place at shallower areas, but this is illustrated just by comparing results in two points in the domain. What I miss is a whole domain picture showing differences in some variable between coupled and uncoupled results to actually confirm that the largest differences occur at shallow places, instead of resulting of a fortunate selection of comparison sites.

According to our interpretation of this point, the current and $Hs$ differences between coupled and uncoupled runs in the whole domain are already shown in Figure 7. This figure shows how the larger effects take place at shallow regions.

The article is centered in wave effects on currents, but might be it would be useful to look at other variables such as sea temperature or salinity, as they might better illustrate the effect of vertical mixing.

The effect of vertical mixing is shown by means of the Brunt–Väisälä frequency, which includes the temperature and salinity information. We have figures with the temperature and salinity evolution during the wind-jet event (see figures below) but we believe that they do not provide new information and it would be redundant. For this reason, we believe that it is better to not show these figures. It would increase the number of figures in the manuscript without giving additional information.

Please also note the supplement to this comment:
https://www.ocean-sci-discuss.net/os-2018-103/os-2018-103-AC2-supplement.pdf

[Figure]

**Temp - uncR**

**Temp - cRS**

**Fig. 1.** Temperature evolution throughout the wind-jet event E3 at point P1.

**Temp - uncR**

**Temp - cRS**

**Fig. 2.** Temperature evolution throughout the wind-jet event E3 at point P3.

**Fig. 3.** Salinity evolution throughout the wind-jet event E3 at point P1.

**Salt - uncR**

**Salt - cRS**

**Fig. 4.** Salinity evolution throughout the wind-jet event E3 at point P3.

**Supplement:**

[Figure]

Figure 1. Temperature evolution throughout the wind-jet event E3 at point P1.

[Figure]

Figure 2. Salinity evolution throughout the wind-jet event E3 at point P1.

[Figure]

Figure 3. Temperature evolution throughout the wind-jet event E3 at point P3.

[Figure]

Figure 4. Salinity evolution throughout the wind-jet event E3 at point P3.